# SEMIR: Semantic Minor-Induced Representation Learning on Graphs for Visual Segmentation

**Luke James Miller** [1]    **Yugyung Lee** [1]

## Abstract

Segmenting small and sparse structures in large-scale images is fundamentally constrained by voxel-level, lattice-bound computation and extreme class imbalance–dense, full-resolution inference scales poorly and forces most pipelines to rely on fixed regionization or downsampling, coupling computational cost to image resolution and attenuating boundary evidence precisely where minority structures are most informative. We introduce **SEMIR** (*Semantic Minor-Induced Representation Learning*), a representation framework that decouples inference from the native grid by learning a task-adapted, topology-preserving latent graph representation with exact decoding. **SEMIR** transforms the underlying grid graph into a compact, boundary-aligned graph minor through parameterized edge contraction, node deletion, and edge deletion, while preserving an exact lifting map from minor predictions to lattice labels. Minor construction is formalized as a few-shot structure learning problem that replaces hand-tuned preprocessing with a *boundary-alignment objective*: minor parameters are learned by maximizing agreement between predicted boundary elements and target-specific semantic edges under a *boundary Dice criterion*, and the induced minor is annotated with scale- and rotation-robust geometric and intensity descriptors and supports efficient region-level inference via message passing on a graph neural network (GNN) with relational edge features. We benchmark **SEMIR** on three tumor segmentation datasets—**BraTS 2021**, **KiTS23**, and **LiTS**—where targets exhibit high structural variability and distributional uncertainty. **SEMIR** yields consistent improvements in *minority-structure Dice* at practical runtime.

[1]Department of Computing, Analytics and Mathematics; University of Missouri-Kansas City, Kansas City, United States. Correspondence to: Luke James Miller <ljmbm5@umsystem.edu>.

*Proceedings of the $43^{rd}$ International Conference on Machine Learning*, Seoul, South Korea. PMLR 306, 2026. Copyright 2026 by the author(s).

More broadly, SEMIR establishes a framework for learning task-adapted, topology-preserving latent representations with exact decoding for high-resolution structured visual data.

## 1. Introduction

Semantic segmentation of medical images presents a fundamental tension between computational scalability and structural fidelity. Volumetric modalities such as CT and MRI routinely produce images exceeding $10^8$ voxels, yet clinically relevant structures—tumors, lesions, fine anatomical boundaries—often occupy fewer than 1% of the volume (Ghankot et al., 2025). Dense, voxel-wise inference scales poorly to such regimes: computational cost grows with image resolution rather than structural complexity, and extreme class imbalance attenuates the gradient signal from minority structures precisely where accurate delineation is most critical (Gao et al., 2025).

Existing approaches address scalability through bottom-up compression: patch-based processing, fixed-grid downsampling, or hand-crafted superpixel extraction. These methods couple the inference resolution to the native lattice or predetermined regionization, sacrificing boundary precision for tractability. The resulting representations are task-agnostic—they do not adapt to the semantic structure of the target domain—and lose boundary evidence before models can leverage it. Multi-class formulations that jointly segment all structures must balance competing objectives across classes with vastly different spatial extent—a trade-off that loss reweighting only partially addresses.

We propose an alternative, top-down formulation: rather than compressing the image into a fixed representation, we *learn* a task-adapted inference space that preserves semantically relevant structure while dramatically reducing computational burden. Our approach, **SEMIR** (*Semantic Minor-Induced Representation Learning*), constructs a compact graph minor $H \preceq G$ from the native voxel lattice $G$ through parameterized edge contraction, node deletion, and edge deletion. The minor $H$ defines a sparse set of supernodes aligned to image boundaries, with node count typically on the order of $\sqrt{|V(G)|}$—reducing a $256^3$ volume

from $\sim 10^7$ voxels to $\sim 10^3$ supernodes while maintaining an exact bijective lifting map for voxel-level prediction. Unlike multi-class pipelines that segment all structures jointly, SEMIR constructs a target-specific inference space: for each structure of interest, we learn a boundary-aligned minor optimized to delineate that structure, reducing the problem to binary classification on the induced graph.

The minor construction is governed by a small parameter set $\Theta$, optimized via black-box few-shot optimization on boundary alignment objective. This few-shot optimization replaces manual threshold tuning with a principled, data-driven procedure: given a handful of labeled examples, the optimizer recovers parameters $\Theta_{\text{opt}}$ such that the induced minor boundaries maximize agreement with ground-truth semantic edges, independent of the downstream label set. The resulting representation generalizes across segmentation tasks sharing similar boundary statistics.

Downstream inference operates entirely on the minor $H$: supernodes are annotated with scale- and rotation-invariant geometric and intensity descriptors, and a graph neural network propagates information across the reduced topology to produce supernode-level predictions. The lifting operator then maps these predictions back to the original lattice, yielding voxel-wise segmentation at a fraction of the computational cost of dense inference.

We evaluate SEMIR on three challenging tumor segmentation benchmarks—BraTS 2021, KiTS23, and LiTS—where targets exhibit high structural variability, extreme class imbalance, and ambiguous boundaries. Across all three datasets, SEMIR yields consistent improvements in minority-structure Dice while maintaining competitive performance on aggregate metrics, demonstrating that *structure-adaptive representations* offer a principled path toward scalable, boundary-aware segmentation in high-resolution medical imaging.

**Contributions** (1) A learned graph minor framework decoupling inference complexity from image resolution with *exact* voxel-level lifting—no interpolation, no boundary artifacts, no approximation. (2) Few-shot, black-box optimization for boundary alignment, replacing manual parameter tuning with a data-driven procedure requiring only 5–20 labeled examples. (3) Scale- and rotation-invariant node/edge descriptors enabling effective message passing on anisotropic medical volumes. (4) Consistent improvements on minority-structure Dice for tumor segmentation—the regime where multi-class methods systematically underperform.

**Conflict of Interest Disclosure.** The authors declare no financial conflicts of interest related to this work.

*Table 1.* Comparison of region reduction approaches. SEMIR provides topology-preserving, boundary-aligned reduction with task-adapted parameters, unlike token merging or fixed pooling methods. [1](Ying et al., 2018) [2](Achanta et al., 2012) [3](Felzenszwalb & Huttenlocher, 2004)

| | Diff-Pool[1] | SLIC[2] | Felzens-walb[3] | **SEMIR** |
|---|---|---|---|---|
| Preserves Topology | $\sim$ | ✓ | ✓ | ✓ |
| Task Aware | ✓ | ✗ | ✗ | ✓ |
| Boundary Aligned | ✗ | $\sim$ | $\sim$ | ✓ |
| Exact Lifting | ✗ | ✓ | ✓ | ✓ |
| Decoding Guarantee | ✗ | ✗ | ✗ | ✓ |

## 2. Related Work

### 2.1. Medical Image Segmentation

U-Net and its variants—incorporating attention, residual connections, and transformer encoders—dominate medical image segmentation (Jiangtao et al., 2025; Luo et al., 2025; Peng et al., 2025). However, computational cost scales with image resolution rather than structural complexity, necessitating patch-based inference or downsampling that discards fine boundary information before the model observes it. The challenge is acute for minority structures: tumors and lesions occupy vanishing fractions of image volume, creating class imbalance that specialized losses only partially mitigate (Yeung et al., 2022; Hosseini, 2025). Boundary-aware and topological losses improve fidelity but remain tied to dense, lattice-bound inference (Zheng et al., 2025).

### 2.2. Superpixel and Oversegmentation Methods

Superpixel algorithms—SLIC (Achanta et al., 2012), Felzenszwalb-Huttenlocher (Felzenszwalb & Huttenlocher, 2004), watershed (Neubert & Protzel, 2014)—reduce complexity by grouping pixels into perceptually coherent regions. However, their regionization is task-agnostic: boundaries follow low-level gradients rather than semantic structure, and parameters require manual tuning. Learned superpixel methods improve boundary adherence but remain constrained by fixed grids or differentiable relaxations (Stutz et al., 2018). In medical imaging, superpixels serve primarily as preprocessing for classical classifiers (Liu et al., 2020). A fundamental limitation is the absence of a principled framework relating induced regions to the original image; mapping predictions back requires heuristics that introduce boundary artifacts. Differentiable pooling methods (MinCutPool, DMoN) learn soft cluster assignments but

lack lifting guarantees; learned superpixel networks (SSN, SEAL) improve boundary adherence but remain grid-bound.

## 2.3. Graph Neural Networks in Medical Imaging

GNNs have been applied to anatomical structure modeling, histopathology cell graphs (Guan et al., 2025), and region adjacency graphs over superpixels (Brussee et al., 2025; Mienye & Viriri, 2025). However, graphs are typically constructed from fixed regionizations—anatomical templates, regular grids, or classical superpixel outputs—determined independently of the downstream task (Ding et al., 2022).

## 2.4. Graph Minors

Graph minors provide a rigorous framework for relating a derived graph $H$ to a parent $G$: $H \preceq G$ if $H$ can be obtained through edge contractions, edge deletions, and node deletions. The Robertson-Seymour theorem establishes that minor-closed properties admit finite forbidden minor characterizations, with polynomial-time testing for fixed $H$ (Robertson & Seymour, 2004; Lovász, 2006). Minors have seen limited vision application. The key insight we exploit is that edge contraction defines a surjection from parent to minor nodes, inducing an exact partition of the original vertex set (Demaine et al., 2005). This partition provides a bijective lifting map for transferring predictions to the original lattice without approximation.

Our approach differs from superpixel methods in three respects: (1) the minor is constructed through parameterized graph operations with formal guarantees; (2) parameters are optimized for boundary alignment via black-box optimization; and (3) the lifting map is exact by construction.

# 3. Method

## 3.1. Overview

SEMIR constructs a compact, boundary-aligned graph minor from the native voxel lattice and performs segmentation via node classification on this reduced representation (Figure 1). The input volume is encoded as a binary tensor $T$ representing the N-connected grid graph $G$. A graph minor is then derived through three parameterized operations: *edge contraction* which merges similar voxels into supernodes, *node deletion* which removes supernodes violating size constraints, and *edge deletion* which severs connections across intensity gradients. The parameter set $\Theta$ defines a family of feasible partitions; few-shot optimization selects from this hypothesis class by maximizing boundary alignment with target-specific semantic edges. Supernodes are annotated with geometric and intensity descriptors; edge features encode scale- and rotation-invariant relative differences. A GNN performs node classification on the graph minor, and

predictions are mapped to the voxel grid via the bijection encoded in $T$ and a lifting function. The minor typically reduces inference from $\sim 10^7$ voxels to $\sim 10^3$ supernodes while guaranteeing exact voxel-level recovery.

## 3.2. Notation

We segment volumetric images $I \in \mathbb{R}^{H \times W \times D \times C}$ into $K$ classes, producing predictions $\hat{Y} \in \{0, \ldots, K-1\}^{H \times W \times D}$. A dataset $\mathcal{D} = \{(I^{(n)}, Y^{(n)})\}_{n=1}^N$ pairs images with ground-truth label maps. The N-connected voxel grid defines graph $G = (V(G), E(G))$ with connectivity $N$. The graph minor $H = (V(H), E(H), X(H), F(H))$ comprises supernodes with feature matrix $X(H) \in \mathbb{R}^{|V(H)| \times d_x}$ and edge features $F(H) \in \mathbb{R}^{|E(H)| \times d_f}$. Few-shot optimization yields $\Theta_{\text{opt}} = R(\mathcal{D}_{\text{few}}, \Theta)$ by aligning predicted boundaries $\hat{Y}_B = S_B(T, \Theta)$ with ground-truth boundaries $Y_B$ relevant to target structures. Full notation table in Appendix A.

## 3.3. Expanded Tensor Representation

The tensor $T \in \{0, \cdots, 255\}^{(2H-1) \times (2W-1) \times (2D-1)}$ interleaves node and edge states along each dimension by encoding a bitflag with the state of the element.

Positions with all even indices $(2j, 2k, 2l)$ encode **nodes** corresponding to voxel $(j, k, l)$. Positions with at least one odd index encode **edges**: an odd index along a single axis indicates a face-adjacent edge (e.g., $(2j+1, 2k, 2l)$ encodes the edge between voxels $(j, k, l)$ and $(j+1, k, l)$), while odd indices along multiple axes indicate diagonal edges under higher connectivity. The exact set of valid edge positions depends on the chosen N-connectivity ($N \in \{6, 10, 18, 26\}$). Each entry is binary:

This representation avoids explicit storage of voxel coordinates and edge lists, reducing memory to $O(|V(G)| + |E(G)|)$ with single-byte entries. Flood-fill visits each edge at most once and terminates immediately upon revisiting assigned voxels, yielding $O(|E(G)| - |E(H)|)$ operations—complexity that scales with contractions performed rather than image resolution. In practice, minor construction on large volumes completes in under one second on CPU using a Rust backend called from Python. Graph-minor construction is performed once after $\Theta_{\text{opt}}$ is fixed, and the GPU receives precomputed graph batches for training and inference; CPU–GPU transfer therefore does not interrupt the forward pass. This makes minor construction negligible compared to downstream inference.

## 3.4. Graph Minor Construction

The initial conversion of an image $I$ to a graph represents each voxel as a node and connects neighboring voxels according to a configurable N-connectivity in 3D, where $N \in \{6, 10, 18, 26\}$ denotes the number of adjacent neigh-

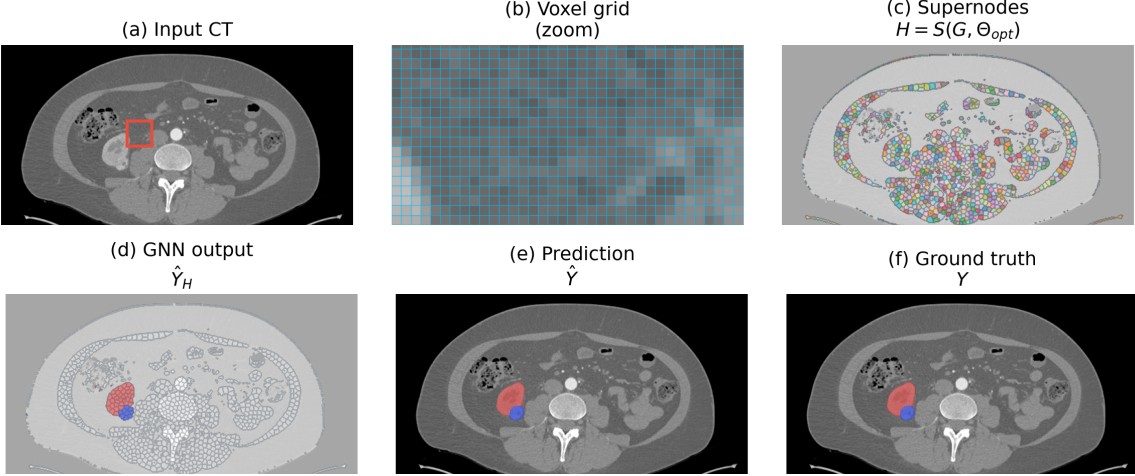

*Figure 1.* SEMIR pipeline visualization on a KiTS23 case. (a) Input contrast-enhanced CT. (b) Zoomed region showing the native voxel grid; each voxel corresponds to a node in $G$. (c) Boundary-aligned graph minor $H \preceq G$ constructed via parameterized edge contraction, node deletion, and edge deletion with few-shot optimized $\Theta_{\text{opt}}$; supernodes colored by ID. (d) Node-level predictions $\hat{Y}_H$ from GNN inference on $H$. (e) Final voxel segmentation $\hat{Y}$ obtained via bijective lifting. (f) Ground truth $Y$. Kidney shown in red, tumor in blue.

bors (face-, face+edge-, face+edge+corner-adjacent, etc.). The choice of $N$ is modality-dependent and ablated in Section 4.4; we default to $N=6$. The resulting grid graph $G = (V(G), E(G))$ is defined as follows:

$$V(G) = \{(j, k, l) : 1 \leq j \leq H, 1 \leq k \leq W, 1 \leq l \leq D\},$$
$$E(G) = (u = (j, k, l), v = (j', k', l')) \in V(G) \times V(G)$$

s.t. $u$ and $v$ are N-connected neighbors.

$$(1)$$

The core data-representation method of this work treats the graph derived from the image (via its expanded tensor $T$) as the parent graph $G$ and constructs a graph minor $H \preceq G$ through a sequence of operations: edge contraction, node deletion, and edge deletion. These operations are performed during a pseudo-random coprime traversal of the nodes to avoid directional bias (e.g., rightward, downward, or backward smearing artifacts in the resulting superpixels).

Edge contraction grows each supernode from a seed voxel. A neighboring voxel $p$ is merged into the current supernode seeded at $s$ when it is connected by an edge in the current graph and satisfies

$$\|I_p - I_s\|_n \leq \psi, \tag{2}$$

where $I_p, I_s \in \mathbb{R}^C$ are voxel intensity vectors, $\|\cdot\|_n$ is a configurable $L_n$ norm (e.g., Euclidean $n = 2$, Manhattan $n = 1$, Chebyshev $n = \infty$), and $\psi$ is the contraction threshold parameter from $\Theta$. Contraction is therefore anchored to the seed voxel rather than to a running supernode mean. This prevents gradual low-contrast transitions from collapsing into a single merged region and instead tends to preserve them as chains of adjacent supernodes.

After contraction, node deletion removes supernodes violating retention criteria. Let $\beta = (\beta_{\min}, \beta_{\max}, m_{\min}, m_{\max})$, where $\beta_{\min}, \beta_{\max}$ control area thresholds and $m_{\min}, m_{\max}$ control intensity thresholds.

$$V_{\text{del}} := \{v \in V(H) : a_v < \beta_{\min} \vee a_v > \beta_{\max}$$
$$\vee \bar{I}_v < m_{\min} \vee \bar{I}_v > m_{\max}\}, \quad (3)$$
$$V(H) \leftarrow V(H) \setminus V_{\text{del}}.$$

This step removes small noisy regions (low area), large background regions (high area), and supernodes with intensities outside the desired range. The lower area bound is specifically intended to suppress acquisition noise: isolated voxels that fail the contraction criterion form singleton or very small supernodes and are pruned before downstream inference. Deleted nodes receive background by default during lifting, so this mechanism is conservative rather than a source of artificial foreground improvement. Finally, edge deletion defines segmentation boundaries by removing edges across strong intensity differences:

$$E_{\text{del}} := \{(v_i, v_j) \in E(H) : \|\bar{I}_{v_i} - \bar{I}_{v_j}\|_n > \alpha\},$$
$$E(H) \leftarrow E(H) \setminus E_{\text{del}}, \tag{4}$$

where $\alpha$ is the edge deletion threshold from $\Theta$. Removed edges separate supernodes across significant intensity gradients, effectively partitioning $H$ into distinct components aligned with image boundaries.

### 3.4.1. FEW-SHOT REPRESENTATIONAL PARAMETER LEARNING

Manual specification of the parameter set $\Theta$ for the graph minor generation function $S(\cdot)$ is labor-intensive and imprecise. Instead, we frame parameter selection as black-box

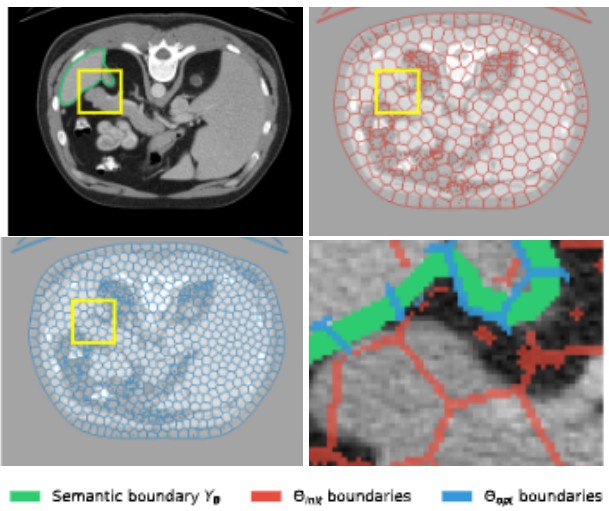

*Figure 2.* Boundary alignment visualization on a LiTS case. (top left) Ground-truth semantic boundary $Y_B$. (top right) Supernode boundaries induced by naive parameters $\Theta_{\text{init}}$. (bottom left) Supernode boundaries induced by optimized $\Theta_{\text{opt}}$. (bottom right) Zoomed comparison. $\Theta_{\text{opt}}$ boundaries align with the semantic edge, while $\Theta_{\text{init}}$ boundaries cut through the tumor boundary.

optimization over the discrete parameter space with a tree-based surrogate $R(\cdot)$ to minimize boundary misalignment on a small held-out subset $\mathcal{D}_{\text{few}} \subset \mathcal{D}$. The optimized parameters are obtained via:

$$\Theta_{\text{opt}} = \arg\min_{\Theta} \ \mathbb{E}_{(I,Y) \sim \mathcal{D}_{\text{few}}} \Big[ L\big(S_B(T, \Theta), Y_B\big) \Big], \quad (5)$$

where $T$ is the expanded tensor representation of $I$, $Y_B$ is the ground-truth binary boundary map derived from the corresponding label map $Y$ (target-specific), and the loss is the inverse Dice-Sørensen coefficient (DSC):

$$L(\hat{Y}_B, Y_B) = 1 - \text{DSC}(\hat{Y}_B, Y_B) = 1 - \frac{2|\hat{Y}_B \cap Y_B|}{|\hat{Y}_B| + |Y_B|}. \quad (6)$$

This boundary-focused loss encourages the minor generation process to produce superpixels whose boundaries match the semantic edges in the ground truth, independent of the specific class identities in $\{0, \dots, K-1\}$. Crucially, $\Theta$ does not configure a fixed segmentation model—it parameterizes a family of graph homomorphisms $\pi_\Theta : G \to H_\Theta$, each inducing a distinct partition of the voxel lattice. Few-shot optimization over $\Theta$ thus constitutes *representation learning over structured latent spaces*: the optimizer selects a partition structure that minimizes semantic boundary risk, rather than tuning hyperparameters within a fixed architecture.

### 3.4.2. GRAPH MINOR FEATURES

Given the optimized parameter set $\Theta_{\text{opt}}$, we execute the graph minor generation $H = S(T, \Theta_{\text{opt}})$, to extract a compact set of node/edge features for downstream prediction.

During the contraction phase of $S(\cdot)$, voxel counts, boundary exposures, coordinates, and intensities are aggregated per supernode. Spatial coordinates are used transiently to compute the covariance matrix of voxel positions, from which the dominant axis and elongation are derived. Intensity statistics (per-channel standard deviation and covariance) are computed directly from the voxel intensities in each supernode. For each supernode $u \in V(H)$ with associated voxels $\mathcal{P}_u = \{p = (j, k, l) : p \text{ belongs to } u\}$ (where $p$ is the voxel coordinate and $I(p) \in \mathbb{R}^C$ its intensity vector), we define the following stored features:

$$a_u := |\mathcal{P}_u|$$
$$\sigma_u := \sqrt{\frac{1}{a_u} \sum_{p \in \mathcal{P}_u} (I(p) - \bar{I}_u) \odot (I(p) - \bar{I}_u)}, \quad (7)$$
$$\Sigma_u := \frac{1}{a_u} \sum_{p \in \mathcal{P}_u} (I(p) - \bar{I}_u)(I(p) - \bar{I}_u)^\top \in \mathbb{R}^{C \times C},$$

$a_u$ is the supernode area/voxel count, $\sigma_u$ is the per-channel standard deviation of intensity, and $\Sigma_u$ is the intensity covariance matrix where $\bar{I}_u = \frac{1}{a_u} \sum_{p \in \mathcal{P}_u} I(p) \in \mathbb{R}^C$ is the mean intensity vector (computed transiently for variance and covariance). The spatial covariance $\Sigma_u^{\text{coord}} \in \mathbb{R}^{3 \times 3}$ is computed transiently from voxel coordinates. Let $\lambda_{u,1} \geq \lambda_{u,2} \geq \lambda_{u,3} \geq 0$ be the eigenvalues of $\Sigma_u^{\text{coord}}$ with corresponding eigenvectors $v_{u,1}, v_{u,2}, v_{u,3}$. We extract:

$$d_u := v_{u,1} \in \mathbb{R}^3,$$
$$\text{elong}_u := \sqrt{(\lambda_{u,1} + \varepsilon)/(\lambda_{u,2} + \varepsilon)}, \quad (8)$$
$$p_u^\star := \arg\min_{(j,k,l) \in \mathcal{P}_u}; \text{lexicographic order}$$

Where $d_u$ is the unit eigenvector of the largest eigenvalue, $\text{elong}_u$ is an elongation ratio with $\varepsilon > 0$ for stability, and $p_u^\star$ is the supernode's canonical voxel. Boundary length, $b_u$, and a 3D compactness proxy (inverse spikiness), $\text{comp}_u$, are computed from the original grid graph $G$ with $\varepsilon > 0$ for numerical stability.

$$b_u := \big|\{(p, q) \in E(G) : p \in \mathcal{P}_u, \ q \notin \mathcal{P}_u\}\big|.$$
$$\text{comp}_u := \frac{36\pi a_u^2}{b_u^3 + \varepsilon} \in (0, 1] \quad (9)$$

While node features capture absolute properties of individual supernodes, edge features encode relative differences between adjacent supernodes, promoting scale- and rotation-invariant representations suitable for the graph neural network. For each edge $e = (u, v) \in E(H)$, we first order the incident supernodes by area, and for any scalar node feature we compute a scale-invariant log-ratio. We calculate relative geometric and orientation edge features for each scalar node feature. Full descriptions can be found in Appendix C.

## 3.5. Downstream Prediction

Given the learned graph minor $H$, we apply a graph neural network for supernode classification. The final voxel-wise segmentation is obtained by lifting these predictions back to the image grid via the bijective mapping in $T$:

$$\hat{Y}_H = \text{GNN}(H), \ \hat{Y} = \text{Lift}(H, \hat{Y}_H, T). \quad (10)$$

where $\hat{Y}_H \in \{0, \ldots, K-1\}^{|V(H)|}$ denotes the predicted class for each supernode $u \in V(H)$. This lifting operation assigns the predicted class of each supernode to all voxels belonging to that supernode, ensuring consistent labeling within each superpixel. During training, the GNN loss and performance metrics are computed by comparing the lifted voxel predictions $\hat{Y}$ against the ground-truth $Y^{(n)}$.

**Composable multi-class inference.** While we train separate binary models per target structure, SEMIR supports compositional multi-class inference by constructing multiple task-adapted minors and resolving overlapping predictions through confidence-weighted voting or energy minimization over the induced partition lattice. This compositional design is intentional: rather than forcing a single representation to balance competing objectives across structures with vastly different spatial extent and boundary statistics, SEMIR constructs a representation optimized for each target. For $K$ target structures, compositional inference requires $K$ minor-construction and GNN passes. The scaling is therefore linear in the number of targets, but each pass operates on the induced graph rather than the full voxel lattice, and the final lifting and overlap-resolution step adds negligible overhead relative to graph construction and GNN inference.

## 3.6. Theoretical Properties

Superpixel segmentation is inherently ill-posed, requiring trade-offs between boundary adherence, compactness, size uniformity, and tractability (Stutz et al., 2018; Wang et al., 2017). Our graph-minor formulation makes these trade-offs explicit and learnable rather than implicit and manual. The parameter set $\Theta = \{\psi, \alpha, \beta\}$ explicitly parameterizes these trade-offs: edge contraction ($\psi$) promotes compact supernodes by merging similar voxels, edge deletion ($\alpha$) enforces boundary separation along intensity gradients, and node deletion ($\beta$) prunes outliers violating size constraints. Crucially, $\Theta$ does not tune a fixed model; it defines the hypothesis class of feasible partitions. Few-shot optimization over $\Theta$ thus constitutes learning the structure of the inference space itself, not merely selecting hyperparameters within a fixed architecture.

**Few-shot generalization of $\Theta$.** The few-shot claim concerns the optimization process, not transfer of a single fixed parameter vector across datasets. The parameter set $\Theta$ is low-dimensional and physically constrained: $\psi$ controls

seed-anchored contraction, $\alpha$ controls boundary separation, and $\beta$ controls size and intensity retention. Thus, the induced family of partitions is much smaller than the class of arbitrary voxel labelings. Few-shot optimization therefore searches over stable boundary statistics rather than class-conditional appearance, which explains why 5–20 labeled examples are sufficient in our experiments while still allowing $\Theta_{\text{opt}}$ to vary across modality, resolution, and target structure.

**Lemma 3.1** (Supernode Connectivity). *For any supernode $u \in V(H)$, the pre-image $\pi^{-1}(u) \subseteq V(G)$ induces a connected subgraph of the N-connected grid $G$.*

*Proof.* Supernodes are constructed via flood-fill from a seed voxel (Algorithm 3). At each iteration, only N-adjacent voxels satisfying the contraction threshold $\psi$ are merged. Since N-adjacency in the grid graph $G$ implies an edge in $E(G)$, and flood-fill proceeds by traversing such edges, the set of merged voxels $\pi^{-1}(u)$ forms a connected subgraph of $G$ by construction. $\square$

**Theorem 3.2** (Lifting Exactness). *Let $H = S(G, \Theta)$ be the induced minor over retained supernodes, with predictions $\hat{Y}_H : V(H) \rightarrow \{0, \ldots, K-1\}$. The lifting operator $\text{Lift}(H, \hat{Y}_H, T)$ assigns labels to voxels exactly: for any $u \in V(H)$ with prediction $\hat{y}_u$, every voxel $v \in \pi^{-1}(u)$ receives label $\hat{y}_u$ with no interpolation or approximation.*

*Proof.* The bijection tensor $T$ encodes the supernode membership of each voxel via the flood-fill assignment. For retained supernodes, $T$ provides a surjective map from voxels to supernodes; lifting inverts this by assigning each voxel the label of its unique containing supernode. No interpolation is required because supernode boundaries are defined exactly by the contraction process, and each voxel belongs to exactly one supernode. $\square$

**Structure-complexity duality.** SEMIR formalizes a fundamental duality: inference cost scales with the topological complexity of semantic boundaries rather than the resolution of the underlying lattice. Dense methods pay for every voxel; SEMIR pays only for structure.

**Additional properties.** Beyond the formal results above, the construction provides: (i) *Adaptivity*—few-shot optimization tunes $\Theta$ to domain-specific boundary statistics without retraining the downstream predictor; (ii) *Invariance*—relative edge features (log-ratios of geometric descriptors, normalized intensity differences) reduce sensitivity to absolute scale and orientation, valuable for anisotropic medical volumes; (iii) *Tractability*—near-linear complexity in voxel count (see Section 3.3).

**Limitations.** The construction lacks the global optimality guarantees of energy-minimization approaches (Kostrykin

*Table 2.* Dataset characteristics.

|            | BraTS     | KiTS        | LiTS      |
|------------|-----------|-------------|-----------|
| Source     | MRI       | CT          | CT        |
| Focus      | Brain     | Kidney      | Liver     |
| Classes    | Glioma    | Tumor/cyst  | Tumor     |
| Trn/Val    | 1,251 / – | 489 / 110   | 131 / 70  |
| Slice size | 240x240   | 512x512     | 512x512   |
| Res. (mm)  | 1x1x1     | variable    | variable  |

& Rohr, 2022). Pseudo-random traversal order introduces stochastic variation across runs. Empirically, variation across runs with different traversals was negligible. Boundary adherence may degrade in low-contrast regions if $\alpha$ is poorly calibrated, and excessive contraction ($\psi$ too permissive) can oversmooth fine structures. Higher N-connectivity (18- or 26-connected) better preserves anisotropic structures at the cost of increased memory. In our implementation, increasing connectivity increases the expanded tensor and edge-state storage, with limited marginal value for volumetric medical images; $N=6$ therefore remains the recommended default for CT and MRI. For thin-structure 2D settings where diagonal adjacency is more important, higher connectivity can be preferable. Ablation studies (Section 4.4) empirically quantify these trade-offs.

# 4. Experiments

## 4.1. Datasets

We evaluate on three volumetric medical imaging benchmarks spanning MRI and CT modalities with varying resolution, anisotropy, and class imbalance characteristics as listed in Table 2. KiTS23 and LiTS use their documented benchmark splits. BraTS 2021 does not provide an official train/validation split; we therefore use an 80/20 train/validation partition, and BraTS comparisons to published methods should be interpreted as contextual rather than split-controlled.

**BraTS 2021** provides multi-parametric MRI at isotropic 1mm resolution for glioma segmentation (WT/TC/ET) (Baid et al., 2021; Bonato et al., 2025). **KiTS23** includes portal venous and nephrogenic CT phases with substantial inter-case variability in slice count and spacing (Heller et al., 2023). **LiTS** exhibits intentionally heterogeneous acquisition parameters, with slice thickness ranging from 0.45–6.0mm (Bilic et al., 2023). Throughout the paper, we abbreviate *BraTS*, *KiTS*, and *LiTS* to indicate these datasets.

## 4.2. Implementation Details

Experiments use an NVIDIA Tesla T4 GPU (16GB) with PyTorch. We train a separate binary model per target structure

(foreground = target, background = all other voxels), reducing each segmentation task to structure-specific inference rather than joint multi-class prediction. This formulation sidesteps class imbalance by construction. See Appendix F for full reproducibility details.

**Architecture and training.** We use a 3-layer GINE (Xu et al., 2019; Hu* et al., 2020) with hidden dimension 128 for supernode prediction. Training uses Adam (Kingma & Ba, 2017) ($lr = 10^{-3}$) for up to 200 epochs with early stopping (patience 10) on validation Dice.

**Parameter optimization.** Minor parameters $\Theta$ are optimized via SMBO with an ExtraTrees surrogate (Bergstra et al., 2011; Berrouachedi et al., 2019) on a few-shot subset ($|\mathcal{D}_{\text{few}}| \in \{5, 10, 20\}$ depending on dataset). Optimized parameters are fixed for all subsequent training; no manual tuning is performed.

## 4.3. Results

We restrict comparisons to methods reporting per-class Dice, as aggregate metrics mask failures on minority structures (cf. Swin UNETR on KiTS: 0.762 aggregate vs. 0.343 tumor). Table 3 provides contextual comparison to published methods under their reported protocols, while Table 4 provides the controlled binary target-vs-rest comparison under identical splits.

Across all three benchmarks, SEMIR consistently improves performance on the clinically relevant minority targets while remaining competitive on the corresponding organ-plus-target aggregates. On BraTS (Table 3), SEMIR attains the strongest performance on TC and ET, indicating improved delineation of tumor core structures that are typically the most fragile under class imbalance and heterogeneous appearance. Notably, gains are concentrated on the more difficult tumor-derived regions rather than being driven solely by the largest whole-tumor mask.

On KiTS and LiTS (Table 3), the same pattern holds: SEMIR yields substantial improvements on the tumor-only class while remaining within the competitive range on organ-plus-tumor aggregates. This is by design: SEMIR does not optimize for aggregate metrics across all classes. Instead, each model targets a single structure, and we report the metric that matters for that target. SEMIR's structure-targeted approach avoids this failure mode. Qualitative examples and ablations in the following sections further characterize where these improvements arise and how they depend on graph construction choices.

SEMIR reduces inference from $\sim 10^7$ to $\sim 10^3$ supernodes—a $10^4\times$ reduction in node count compared to dense methods (Appendix E). This complexity scales with image structure rather than resolution, enabling efficient processing of high-resolution volumes.

*Table 3.* Comparison across BraTS 2021, KiTS23, and LiTS benchmarks. Key: **Best**, *Second.* BraTS regions: TC (Tumor Core), WT (Whole Tumor), ET (Enhancing Tumor). KiTS regions: KTC (Kidney+Tumor+Cyst), TC (Tumor+Cyst), T (Tumor). LiTS regions: LT (Liver+Tumor), T (Tumor).
[1](El Badaoui et al., 2025) [2](Tobias et al., 2025) [3](Alwadee et al., 2025) [4](Bonato et al., 2025) [5](Nowakowski & Patel, 2025) [6](Zhang et al., 2025a) [7](Alonso-Monsalve et al., 2025) [8](Uhm et al., 2023) [9](Liu et al., 2025) [10](Yang et al., 2025) [11](Ren et al., 2025) [12](Zheng et al., 2025) [13](Zhang et al., 2025b) [14](Saifullah & Dreżewski, 2025) [15](Perera et al., 2024) [16](Lilhore et al., 2025) [17](Lei et al., 2022) [18](Li et al., 2018) [19](Myronenko et al., 2024) [20](Kaczmarska & Majek, 2024) [21](Pandey et al., 2024) [22](Liao et al., 2024) [23](xmed lab, 2024)

| **BraTS 2021** | | | | **KiTS23** | | | | **LiTS** | | |
| **Method** | TC | WT | ET | **Method** | KTC | TC | T | **Method** | LT | T |
|---|---|---|---|---|---|---|---|---|---|---|
| Swin UNETR[1] | 0.682 | 0.823 | 0.748 | ConvOccNet[5] | 0.943 | 0.725 | 0.693 | Yang et al.[10] | 0.971 | 0.721 |
| 3DCATBraTS[1] | 0.784 | 0.851 | 0.792 | AlignUNet[6] | 0.781 | —— | 0.613 | MA-UNet[10] | 0.960 | 0.729 |
| E-CATBraTS[1] | 0.726 | 0.802 | 0.781 | Alonso[7] | **0.958** | *0.856* | 0.803 | Lgma-net[11] | **0.977** | *0.874* |
| PAU-Net[2] | 0.879 | 0.915 | 0.835 | 3DU-Net Ens.[8] | *0.948* | 0.776 | 0.738 | DefED-Net[17] | 0.963 | *0.875* |
| LATUP-Net[3] | 0.895 | 0.902 | 0.839 | Swin UNETR[20] | 0.762 | 0.439 | 0.343 | Swin-UNet[12] | 0.967 | 0.857 |
| Ext. nnUNet[4] | 0.878 | 0.927 | 0.845 | Multi-Planner[21] | 0.831 | 0.130 | 0.260 | Li & Zhao[12] | 0.967 | 0.865 |
| *Transformer / State-Space* | | | | *Challenge Winners* | | | | *Transformer / Hybrid* | | |
| GTMamba[13] | *0.940* | *0.943* | 0.884 | Auto3DSeg[19] | 0.926 | 0.784 | 0.751 | HDenseUNet[18] | 0.961 | 0.722 |
| PSO-UNet[14] | —— | **0.958** | —— | Uhm et al.[8] | *0.948* | 0.776 | 0.738 | MAcGAN[23] | 0.970 | 0.785 |
| SegFormer[15] | 0.822 | 0.899 | 0.742 | | | | | SwinUNETR[23] | 0.867 | 0.742 |
| MS3D-CNN[16] | 0.912 | 0.928 | 0.857 | | | | | MedNeXtL[23] | 0.946 | 0.779 |
| **SEMIR (ours)** | **0.941** | 0.920 | **0.894** | **SEMIR** | 0.945 | **0.861** | **0.819** | **SEMIR** | 0.959 | **0.891** |

*Table 4.* Controlled binary target-vs-rest comparison against nnU-Net under identical splits. Runtime reports end-to-end training time in the practical hardware regime used for each method. †SEMIR completed on a 16GB T4 GPU; nnU-Net required A100-class hardware to complete in practical time.

| | | **nnU-Net** | **SEMIR** | |
| **Dataset** | **Target** | **DSC** | **DSC** | **Time** |
|---|---|---|---|---|
| BraTS | ET | 0.812 | 0.894±.006 | 43h/2.5h† |
| BraTS | TC | 0.829 | 0.941±.002 | 39h/1.6h† |
| KiTS | T | 0.720 | 0.819±.006 | 19h/0.8h† |
| LiTS | T | 0.733 | 0.891±.007 | 11h/0.6h† |

*Table 5.* Ablation summary. BraTS ET Dice and NWPU VHR-10 IoU.

| | **Ablation** | **BraTS ET** | **NWPU** |
|---|---|---|---|
| | Full SEMIR | **0.894** | **0.862** |
| *Minor operations* | No edge contraction | 0.441 | 0.408 |
| | No edge deletion | 0.719 | 0.681 |
| | No node deletion | 0.812 | 0.749 |
| *Parameter optimization* | Learned ($|\mathcal{D}_{\text{few}}|$=5) | 0.894 | 0.789 |
| | Fixed (best manual) | 0.837 | 0.763 |
| *Features* | No edge features | 0.725 | 0.741 |
| | No spatial features | 0.661 | 0.629 |

Ablation studies (Section 4.4) confirm that our method transfers to non-medical domains: on NWPU VHR-10 aerial imagery, the learned minor achieves 0.862 small-object IoU, demonstrating that boundary-aligned minor construction is not specific to volumetric medical data.

Figures 1 and 2 visualize SEMIR's representation learning. The pipeline figure shows how the boundary-aligned minor $H$ reduces a KiTS volume to a sparse supernode graph while preserving structure for accurate tumor delineation. The boundary alignment figure demonstrates the effect of few-shot optimization: $\Theta_{\text{opt}}$ produces supernodes whose boundaries track semantic edges, whereas naive parameters $\Theta_{\text{init}}$ cut arbitrarily through the tumor boundary.

## 4.4. Ablation Studies

We ablate key components on BraTS (minority classes: ET, TC) and a 100-sample subset of NWPU VHR-10 (Cheng et al., 2014), an aerial detection benchmark, to verify that SEMIR's minor construction generalizes beyond medical imaging to 2D RGB imagery with small, sparse targets. Full results in Appendix D.

**Key findings.** Disabling edge contraction causes severe fragmentation (ET Dice drops 51%). Few-shot optimization outperforms manual tuning even with 5 samples. Spatial features (compactness, elongation) are essential for irregular structures, with 24–27% degradation when removed. Higher N-connectivity (18 vs. 6) improves small-object IoU by 3–5% at 2.5× runtime cost; $L_{\infty}$ norm shows strongest robustness on multi-channel data.

# 5. Conclusion

We introduced SEMIR, a framework that constructs a learned graph minor over the voxel lattice, enabling segmentation inference on a compact, boundary-aligned representation with exact lifting back to voxel-level predictions. Few-shot optimization of minor construction parameters replaces manual superpixel tuning with a principled, data-driven procedure. Across three tumor segmentation benchmarks, SEMIR yields consistent improvements on structure-specific Dice for clinically critical targets—the regime where multi-class methods suffer from imbalance-induced gradient attenuation. By reducing each task to binary segmentation on a target-adapted graph, SEMIR sidesteps the balancing problem entirely—while reducing inference to 1–3% of the original node count.

Limitations include sensitivity to boundary statistics in the few-shot set and evaluation restricted to volumetric CT/MRI. The current formulation decouples minor construction from downstream prediction; this modularity means the learned minor can serve as a preprocessing stage for any downstream model (CNNs, transformers, GNNs), though end-to-end joint optimization remains an obvious target for future work. Extension to additional modalities (histopathology, ultrasound) and theoretical analysis of minor structure under varying acquisition conditions warrant further exploration.

# Impact Statement

This work aims to improve segmentation of minority structures in medical images, with potential applications in tumor detection and treatment planning. Two considerations merit attention: (1) the benchmarks used do not represent a global sample; performance on underrepresented populations requires further validation. (2) SEMIR's node deletion mechanism excludes image regions outside learned thresholds, which improves efficiency but could discard atypical pathologies. Clinical implementations should maintain radiologist oversight and preserve access to full-resolution data. We believe improved minority-structure segmentation offers meaningful clinical benefit when deployed with appropriate validation and safeguards.

# Acknowledgments

Luke Miller and Yugyung Lee acknowledge support from the National Science Foundation (NSF) under Award No. 2152057.

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

# A. Notation

| Symbol | Description |
| --- | --- |
| $I \in \mathbb{R}^{H \times W \times D \times C}$ | Input medical image volume (height $\times$ width $\times$ depth $\times$ channels) |
| $I_{j,k,l} \in \mathbb{R}^{C}$ | Intensity vector at voxel coordinate $(j, k, l)$ |
| $Y \in \{0, \ldots, K-1\}^{H \times W \times D}$ | Ground-truth voxel class labels |
| $Y_B \in \{0, 1\}^{H \times W \times D}$ | Binary ground-truth boundary map |
| $\hat{Y} \in \{0, \ldots, K-1\}^{H \times W \times D}$ | Predicted voxel class map |
| $T \in \{0, 1\}^{(2H-1) \times (2W-1) \times (2D-1)}$ | Expanded binary tensor encoding graph nodes and edges |
| $G = (V(G), E(G))$ | Original N-connected grid graph of $I$ ($N \in \{6, 10, 18, 26\}$) |
| $H = (V(H), E(H), X(H), F(H))$ | Graph minor of $G$ |
| $X(H) \in \mathbb{R}^{d_x \times |V(H)|}$ | Node feature matrix |
| $a_u$ | Area: voxel count of supernode $u$ |
| $b_u$ | Boundary length: number of exposed edges of supernode $u$ |
| $\text{comp}_u = \frac{36\pi a_u^2}{b_u^3 + \varepsilon}$ | 3D compactness of supernode $u$ ($\varepsilon > 0$ for stability) |
| $d_u \in \mathbb{R}^3$ | Dominant axis: unit eigenvector of largest eigenvalue of spatial covariance |
| $\text{elong}_u$ | Elongation ratio: derived from eigenvalues of spatial covariance |
| $p_u^\star$ | Canonical voxel: lexicographically smallest coordinate in supernode $u$ |
| $\sigma_u \in \mathbb{R}^C$ | Per-channel standard deviation of intensities in supernode $u$ |
| $\Sigma_u \in \mathbb{R}^{C \times C}$ | Covariance matrix of intensity vectors in supernode $u$ |
| $F(H) \in \mathbb{R}^{d_f \times |E(H)|}$ | Edge feature matrix (log-ratios of corresponding node features) |
| $\Theta = \{\psi, \alpha, \beta\}$ | Parameters for graph minor generation $S(T, \Theta)$ |
| $S(T, \Theta) \mapsto (T, H)$ | Graph minor generation function |
| $S_B(T, \Theta) \mapsto \hat{Y}_B$ | Boundary voxel map extractor |
| $R(\mathcal{D}_{\text{few}}, \Theta)$ | Few-shot parameter optimization |
| $\mathcal{D}_{\text{few}} \subset \mathcal{D}$ | Few-shot tuning subset of dataset |
| $\mathcal{D} = \{(I^{(n)}, Y^{(n)})\}_{n=1}^N$ | Full dataset |
| $\text{Lift}(H, \hat{Y}_H, T) \mapsto \hat{Y}$ | Mapping of supernode predictions $\hat{Y}_H$ back to voxel grid |

# B. Algorithms

---

**Algorithm 1** GETCOPRIME: Find coprime step for pseudo-random traversal

---

**Require:** Dimension size $n$, divisor $d$
**Ensure:** Step size $s$ coprime to $n$
  $s \leftarrow \lfloor n / \max(d, 1) \rfloor$
  $s \leftarrow \text{clamp}(s, 1, n-1)$
  **for** $i = s$ to $n - 1$ **do**
    **if** $\gcd(n, i) = 1$ **then**
      **Return** $i$
    **end if**
  **end for**
  **for** $i = s - 1$ to $1$ **do**
    **if** $\gcd(n, i) = 1$ **then**
      **Return** $i$
    **end if**
  **end for**
  **Return** $1$

---

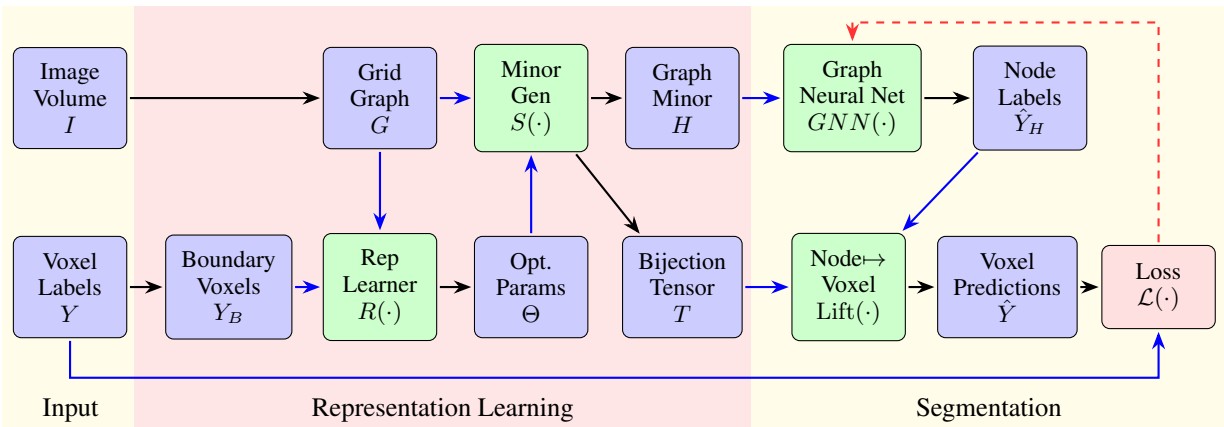

*Figure 3.* SEMIR pipeline overview. From the input volume $I$, a boundary-aware graph minor $H$ is constructed using few-shot-optimized parameters $\Theta$. A graph neural network yields supernode predictions $\hat{Y}_H$, which are bijectively lifted via tensor $T$ to produce the final voxel segmentation $\hat{Y}$. Ground-truth voxel labels $Y$ provide supervision through the loss and few-shot boundary alignment.

---

**Algorithm 2** MINORCONSTRUCTION: Graph minor construction $S(I, \Theta)$

---

**Require:** Image volume $I \in \mathbb{R}^{H \times W \times D \times C}$, parameters $\Theta = \{\psi, \alpha, \beta\}$
**Ensure:** Tensor $T$, graph minor $H = (V, E, X, F)$
  {Initialize expanded tensor}
  $T \leftarrow \mathbf{1}^{(2H-1) \times (2W-1) \times (2D-1)}$
  {Compute coprime steps for pseudo-random traversal}
  $(s_h, s_w, s_d) \leftarrow (\text{GETCOPRIME}(H, d), \text{ GETCOPRIME}(W, d), \text{ GETCOPRIME}(D, d))$
  $(r_0, c_0, l_0) \leftarrow$ random start position
  $V \leftarrow \emptyset, X \leftarrow \emptyset$
  {Phase 1: Edge contraction via flood-fill with node deletion}
  **for** $i = 0$ to $H - 1$ **do**
    **for** $j = 0$ to $W - 1$ **do**
      **for** $k = 0$ to $D - 1$ **do**
        $(r, c, l) \leftarrow ((r_0 + i \cdot s_h) \bmod H, (c_0 + j \cdot s_w) \bmod W, (l_0 + k \cdot s_d) \bmod D)$
        **if** $T[2r, 2c, 2l]$ is visited **then**
          **continue**
        **end if**
        $(T, x_u, \mathcal{P}_u) \leftarrow \text{FLOODFILLCONTRACT}(I, T, (r, c, l), \psi, \alpha)$
        {Node deletion check}
        **if** $\beta_{\min} < |\mathcal{P}_u| < \beta_{\max}$ **then**
          $V \leftarrow V \cup \{u\}, \quad X \leftarrow X \cup \{x_u\}$
        **else**
          Mark all voxels in $\mathcal{P}_u$ as deleted in $T$
        **end if**
      **end for**
    **end for**
  **end for**
  {Phase 2: Build edge list and edge features from surviving nodes}
  $(E, F) \leftarrow \text{EXTRACTEDGES}(T, V, X)$
  **Return** $(T, H = (V, E, X, F))$

---

---

**Algorithm 3** FLOODFILLCONTRACT: Region growing with contraction and edge deletion

---

**Require:** Image $I$, tensor $T$, seed voxel $(r, c, l)$, merge threshold $\psi$, cut threshold $\alpha$
**Ensure:** Updated $T$, node features $x_u$, voxel set $\mathcal{P}_u$

  $v_{\text{seed}} \leftarrow I[r, c, l]$ {Seed intensity (canonical voxel)}
  Initialize stack $S \leftarrow \{(2r, 2c, 2l)\}$, voxel set $\mathcal{P}_u \leftarrow \emptyset$
  Initialize running statistics: area, sums for coordinates, intensities, products
  **while** $S \neq \emptyset$ **do**
    Pop $(y, x, z)$ from $S$
    **if** $T[y, x, z]$ is visited **then**
      **continue**
    **end if**
    Mark $T[y, x, z]$ as visited
    $(r', c', l') \leftarrow (y/2, x/2, z/2)$ {Original voxel coordinates}
    $\mathcal{P}_u \leftarrow \mathcal{P}_u \cup \{(r', c', l')\}$
    Update running statistics with $(r', c', l')$ and $I[r', c', l']$
    Update canonical voxel if $(r', c', l')$ is lexicographically smaller
    **for** each N-connected neighbor direction $\delta$ **do**
      $(y', x', z') \leftarrow (y, x, z) + 2\delta$ {Neighbor node position}
      **if** $(y', x', z')$ out of bounds **or** $T[y', x', z']$ is merged **then**
        **continue**
      **end if**
      $\text{diff} \leftarrow \|v_{\text{seed}} - I[y'/2, x'/2, z'/2]\|_n$
      **if** $\text{diff} \leq \psi$ **then**
        Mark $T[y', x', z']$ as merged
        Push $(y', x', z')$ onto $S$
      **else**
        **if** $\text{diff} \geq \alpha$ **then**
          $(e_y, e_x, e_z) \leftarrow (y, x, z) + \delta$ {Edge position}
          Mark $T[e_y, e_x, e_z]$ as edge-deleted
        **end if**
        Mark $T[y, x, z]$ as boundary-adjacent
      **end if**
    **end for**
    **if** $T[y, x, z]$ is boundary-adjacent **then**
      Increment boundary length counter
    **end if**
  **end while**
  $x_u \leftarrow$ COMPUTENODEFEATURES(running statistics, $\mathcal{P}_u$)
  **Return** $(T, x_u, \mathcal{P}_u)$

---

---

**Algorithm 4** EXTRACTEDGES: Build edge list and features from tensor

---

**Require:** Tensor $T$, node set $V$, node features $X$
**Ensure:** Edge set $E$, edge features $F$
  $E \leftarrow \emptyset$, $F \leftarrow \emptyset$
  **for** each node $u \in V$ **do**
    $(r, c, l) \leftarrow$ canonical voxel of $u$
    **for** each N-connected neighbor direction $\delta$ **do**
      $(e_y, e_x, e_z) \leftarrow (2r, 2c, 2l) + \delta$ {Edge position}
      **if** $T[e_y, e_x, e_z]$ is not edge-deleted **then**
        $(y', x', z') \leftarrow (2r, 2c, 2l) + 2\delta$ {Neighbor node position}
        $v \leftarrow$ node containing voxel $(y'/2, x'/2, z'/2)$
        **if** $v \in V$ and $v \neq u$ and $(u, v) \notin E$ **then**
          $E \leftarrow E \cup \{(u, v)\}$
          $f_{uv} \leftarrow$ COMPUTEEDGEFEATURES$(X_u, X_v)$
          $F \leftarrow F \cup \{f_{uv}\}$
        **end if**
      **end if**
    **end for**
  **end for**
  **Return** $(E, F)$

---

**Algorithm 5** FEWSHOTOPTIMIZATION: Parameter optimization $R(\mathcal{D}_{\text{few}}, \Theta)$

---

**Require:** Few-shot dataset $\mathcal{D}_{\text{few}}$, parameter bounds $\Theta$, iterations $N_{\text{iter}}$, initial samples $N_{\text{init}}$
**Ensure:** Optimized parameters $\Theta_{\text{opt}}$
  {Initialize surrogate with random samples}
  $\mathcal{H} \leftarrow \emptyset$ {History of $(\theta, \text{loss})$ pairs}
  **for** $i = 1$ to $N_{\text{init}}$ **do**
    $\theta \sim \text{Uniform}(\Theta)$
    $\mathcal{L} \leftarrow$ EVALUATEBOUNDARYLOSS$(\theta, \mathcal{D}_{\text{few}})$
    $\mathcal{H} \leftarrow \mathcal{H} \cup \{(\theta, \mathcal{L})\}$
  **end for**
  {Sequential model-based optimization}
  **for** $i = 1$ to $N_{\text{iter}}$ **do**
    Fit ExtraTrees surrogate $\hat{f}$ on $\mathcal{H}$
    $\theta_{\text{next}} \leftarrow \arg\max_\theta \text{EI}(\theta; \hat{f}, \mathcal{H})$ {Expected Improvement}
    $\mathcal{L} \leftarrow$ EVALUATEBOUNDARYLOSS$(\theta_{\text{next}}, \mathcal{D}_{\text{few}})$
    $\mathcal{H} \leftarrow \mathcal{H} \cup \{(\theta_{\text{next}}, \mathcal{L})\}$
  **end for**
  $\Theta_{\text{opt}} \leftarrow \arg\min_{(\theta, \mathcal{L}) \in \mathcal{H}} \mathcal{L}$
  **Return** $\Theta_{\text{opt}}$

---

---

**Algorithm 6** EVALUATEBOUNDARYLOSS: Compute boundary alignment loss

---

**Require:** Parameters $\theta$, few-shot dataset $\mathcal{D}_{\text{few}}$
**Ensure:** Mean boundary loss $\bar{\mathcal{L}}$
   $\mathcal{L}_{\text{total}} \leftarrow 0$
   **for** each $(I, Y) \in \mathcal{D}_{\text{few}}$ **do**
      $Y_B \leftarrow$ EXTRACTGROUNDTRUTHBOUNDARY$(Y)$ {Voxels adjacent to different class}
      $(T, \_) \leftarrow$ MINORCONSTRUCTION$(I, \theta)$ {Features not needed}
      $\hat{Y}_B \leftarrow$ EXTRACTMINORBOUNDARY$(T)$ {Boundary-adjacent voxels in $T$}
      $\mathcal{L}_{\text{total}} \leftarrow \mathcal{L}_{\text{total}} + (1 - \text{DSC}(\hat{Y}_B, Y_B))$
   **end for**
   **Return** $\bar{\mathcal{L}} \leftarrow \mathcal{L}_{\text{total}}/|\mathcal{D}_{\text{few}}|$

---

---

**Algorithm 7** LIFT: Map supernode predictions to voxel grid

---

**Require:** Tensor $T$, node predictions $\hat{Y}_H$, node set $V$ with canonical voxels
**Ensure:** Voxel predictions $\hat{Y} \in \{0, \ldots, K-1\}^{H \times W \times D}$
   $\hat{Y} \leftarrow \mathbf{0}^{H \times W \times D}$ {Background default for deleted nodes}
   **for** each node $u \in V$ **do**
      $(r, c, l) \leftarrow$ canonical voxel of $u$
      Initialize stack $S \leftarrow \{(2r, 2c, 2l)\}$, visited set $\mathcal{V} \leftarrow \emptyset$
      **while** $S \neq \emptyset$ **do**
         Pop $(y, x, z)$ from $S$
         **if** $(y, x, z) \in \mathcal{V}$ **then**
            **continue**
         **end if**
         $\mathcal{V} \leftarrow \mathcal{V} \cup \{(y, x, z)\}$
         $\hat{Y}[y/2, x/2, z/2] \leftarrow \hat{Y}_H[u]$
         **for** each N-connected neighbor direction $\delta$ **do**
            $(e_y, e_x, e_z) \leftarrow (y, x, z) + \delta$ {Edge position}
            $(y', x', z') \leftarrow (y, x, z) + 2\delta$ {Neighbor node position}
            **if** in bounds **and** $T[e_y, e_x, e_z]$ not edge-deleted **and** $T[y', x', z']$ is merged **then**
               Push $(y', x', z')$ onto $S$
            **end if**
         **end for**
      **end while**
   **end for**
   **Return** $\hat{Y}$

---

## C. Full Description of Features

While node features capture absolute properties of individual supernodes, edge features encode relative differences between adjacent supernodes, promoting scale- and rotation-invariant representations suitable for the graph neural network.

For each edge $e = (u, v) \in E(H)$, we first order the incident supernodes by area:

$$u^+ := \arg\max\{a_u, a_v\}, \qquad u^- := \arg\min\{a_u, a_v\}. \tag{11}$$

For any scalar node feature $g(\cdot)$ (e.g., $a_u$, $b_u$, $\text{comp}_u$, $\text{elong}_u$), we compute a scale-invariant log-ratio:

$$r_g(e) := \log \frac{g(u^+) + \varepsilon}{g(u^-) + \varepsilon}, \tag{12}$$

with $\varepsilon > 0$ for numerical stability.

We calculate relative geometric and orientation edge features for each scalar node feature.

$$\Delta_{\mu}(e) := \frac{\mu_u - \mu_v}{\sqrt{\lambda_{u,1} + \lambda_{v,1} + \varepsilon}} \in \mathbb{R}^3, \tag{13}$$

$$\cos\theta(e) := |d_u^\top d_v| \in [0, 1], \tag{14}$$

where $\mu_u$ and $\mu_v$ are the (transiently computed) centroids of supernodes $u$ and $v$, $\lambda_{u,1}$ and $\lambda_{v,1}$ are their respective largest spatial eigenvalues, and $d_u, d_v$ are the dominant axes.

For intensity-based features, we use normalized per-channel differences:

$$\Delta_I(e) := \frac{\bar{I}_u - \bar{I}_v}{\sqrt{s_{I,u}^2 + s_{I,v}^2 + \varepsilon}} \in \mathbb{R}^C, \tag{15}$$

where $\bar{I}_u, \bar{I}_v$ are the mean intensity vectors and $s_{I,u}^2, s_{I,v}^2$ are the per-channel variance vectors (i.e., element-wise squares of $\sigma_u, \sigma_v$).

These relative features ($r_g(e)$ for relevant scalars, $\Delta_{\mu}(e)$, $\cos\theta(e)$, $\Delta_I(e)$) are stored in the edge feature matrix $F(H)$.

## D. Extended Ablation Studies

### D.1. Graph-Minor Construction Operations

Table 6 shows the impact of disabling individual operations. Edge contraction prevents severe fragmentation; edge deletion separates minority structures; node deletion prunes noise and oversized background supernodes. Uniform grid downsampling (intensity-agnostic) serves as a weak baseline.

*Table 6.* Ablation of graph-minor construction operations.

| | BraTS | | NWPU |
| Variant | ET | TC | VHR-10 |
|---|---|---|---|
| Full SEMIR (learned $\Theta$) | **0.894** | **0.941** | **0.862** |
| No edge contraction | 0.441 | 0.492 | 0.408 |
| No edge deletion | 0.719 | 0.774 | 0.681 |
| No node deletion | 0.812 | 0.837 | 0.749 |
| Grid downsampling | 0.393 | 0.351 | 0.252 |

### D.2. Few-Shot Parameter Optimization

Table 7 compares learned $\Theta$ against fixed thresholds across few-shot set sizes. Performance saturates around 5 samples for BraTS and 20 samples for multi-channel NWPU.

*Table 7.* Few-shot optimization strategy and set size $|\mathcal{D}_{\text{few}}|$.

| $\Theta$ strategy | $|\mathcal{D}_{\text{few}}|$ | BraTS ET | NWPU |
|---|---|---|---|
| Learned | 1 | 0.766 | 0.641 |
| Learned | 5 | **0.894** | 0.789 |
| Learned | 10 | 0.891 | 0.854 |
| Learned | 20 | 0.890 | **0.862** |
| Fixed (loose) | – | 0.499 | 0.544 |
| Fixed (tight) | – | 0.837 | 0.749 |
| Fixed (visual) | – | 0.711 | 0.763 |

### D.3. Distance Norms and Connectivity

Table 8 evaluates distance norms and N-connectivity. Norms are equivalent on single-channel BraTS. On multi-channel NWPU, $L_\infty$ shows strongest robustness to channel-specific outliers; higher connectivity improves small-object IoU at moderate runtime cost.

*Table 8.* Distance norm and N-connectivity ablation.

| Norm | N | BraTS ET | | NWPU | |
|---|---|---|---|---|---|
| | | DSC | Time | IoU | Time |
| $L_1$ | 6 | 0.894 | 1.0× | 0.812 | 1.0× |
| $L_1$ | 10 | 0.889 | 1.6× | 0.819 | 1.8× |
| $L_1$ | 18 | 0.890 | 2.7× | 0.825 | 2.3× |
| $L_2$ | 6 | — | — | 0.796 | 1.4× |
| $L_2$ | 10 | — | — | 0.804 | 2.5× |
| $L_2$ | 18 | — | — | 0.813 | 3.8× |
| $L_\infty$ | 6 | — | — | 0.809 | 1.1× |
| $L_\infty$ | 10 | — | — | 0.836 | 1.9× |
| $L_\infty$ | 18 | — | — | **0.849** | 2.5× |

### D.4. Feature Design

Removing relative edge features causes ET Dice to drop by 11–19% on BraTS and small-object IoU by 7–14% on NWPU, underscoring their importance for scale- and rotation-invariant relational modeling. Disabling spatial features (compactness, elongation, dominant axis) reduces performance by 24–27% in both domains, with greater impact on irregularly shaped objects.

## E. Complexity Analysis

**Minor construction.** Each voxel is visited exactly once during flood-fill traversal, and each edge is examined at most twice. Construction is therefore $O(N \cdot HWD)$ where $N \in \{6, 10, 18, 26\}$ is the grid connectivity—linear in voxel count regardless of the induced minor size.

**GNN inference.** Message passing operates on the minor $H$, with cost $O(L(|V(H)| + |E(H)|))$ for $L$ layers. The critical quantity is $|V(H)|$, which depends on image content rather than resolution: $|V(H)|$ reflects the number of intensity-homogeneous regions satisfying the learned thresholds $\Theta_{\text{opt}}$.

In the degenerate case (all adjacent voxels differ by more than $\psi$), $|V(H)| = O(HWD)$. In practice, medical images contain large homogeneous regions (background, parenchyma) and $|V(H)| \ll HWD$. Table 9 reports empirical supernode counts.

*Table 9.* Empirical supernode counts $|V(H)|$ across benchmarks.

| | BraTS | KiTS | LiTS |
|---|---|---|---|
| Volume size | $240 \times 240 \times 155$ | variable | variable |
| Voxels $|V(G)|$ | $\sim 8.9 \times 10^6$ | $\sim 10^7$–$10^8$ | $\sim 10^7$–$10^8$ |
| Supernodes $|V(H)|$ | $2034 \pm 187$ | $1429 \pm 456$ | $1075 \pm 297$ |
| Reduction factor | $\sim 4400\times$ | $\sim 10^4\times$ | $\sim 10^4\times$ |

Table 10 compares the number of inference units across segmentation paradigms. Dense methods (U-Net variants, transformers) perform per-voxel prediction, scaling directly with image resolution regardless of structural complexity. Patch-based inference reduces peak memory but does not reduce total computation, as overlapping windows collectively cover all voxels. Classical superpixel methods (SLIC, Felzenszwalb) reduce inference units but use task-agnostic regionization with manually tuned parameters.

SEMIR achieves 3–4 orders of magnitude reduction in inference nodes compared to dense methods by learning a boundary-aligned graph minor adapted to the target structure. Empirical supernode counts from Table 9 confirm this reduction: BraTS volumes ($\sim 8.9 \times 10^6$ voxels) yield $2034 \pm 187$ supernodes; KiTS and LiTS volumes ($\sim 10^7$–$10^8$ voxels) yield 1075–1429 supernodes. The downstream GNN operates entirely on this reduced representation, with voxel-level predictions recovered via exact lifting (Theorem 3.2).

*Table 10.* Inference complexity across segmentation paradigms for a typical medical volume ($256 \times 256 \times 128$ voxels, $\sim 8.4 \times 10^6$ total). SEMIR operates on 3–4 orders of magnitude fewer nodes than dense methods while maintaining exact voxel-level output via lifting.

| Method | Paradigm | Inference Units | Unit Type |
|---|---|---|---|
| 3D U-Net | Dense | $8.4 \times 10^6$ | voxels |
| nnU-Net | Dense | $8.4 \times 10^6$ | voxels |
| Swin UNETR | Dense + Attention | $8.4 \times 10^6$ | voxels |
| TransUNet | Patch + Dense | $8.4 \times 10^6$ | voxels |
| Patch-based ($64^3$) | Sliding window | $8.4 \times 10^6$ | voxels$^\dagger$ |
| Patch-based ($128^3$) | Sliding window | $8.4 \times 10^6$ | voxels$^\dagger$ |
| SLIC superpixels | Fixed regions | $10^4$–$10^5$ | superpixels |
| Felzenszwalb | Fixed regions | $10^4$–$10^5$ | superpixels |
| **SEMIR (ours)** | Learned minor | $10^3$–$10^4$ | supernodes |

$^\dagger$Patch-based methods reduce memory per forward pass but process equivalent total voxels with overlapping windows.

# F. Implementation Details

**Hardware and software.** All experiments were conducted in Google Colab on an NVIDIA Tesla T4 GPU (16GB). Preprocessing and learning are implemented in Python using NumPy and PyTorch.

**GNN architecture.** We use a 3-layer GIN variant with edge features (GINE) (Xu et al., 2019; Hu* et al., 2020). Each layer applies message passing with edge-conditioned messages; node and edge features are embedded by MLPs with ReLU activations and hidden dimension $d = 128$. A final per-node linear layer outputs logits for each supernode.

**Training details.** We train with Adam (Kingma & Ba, 2017) using learning rate $10^{-3}$ and no weight decay for up to 200 epochs. Early stopping monitors validation Dice on the binarized foreground target with patience 10. Batch size is 1–4 volumes depending on memory constraints from variable graph sizes.

**Few-shot dataset sizes.** We use $|\mathcal{D}_{\text{few}}| = 5$ for BraTS, 10 for KiTS, and 20 for LiTS, drawn from the training split.

**Graph construction settings.** We evaluate N-connectivity $N \in \{6, 10, 18\}$ in the expanded tensor $T$, using $N = 6$ by default. For intensity-distance computations, we evaluate $L_1$ (Manhattan), $L_2$ (Euclidean), and $L_\infty$ (Chebyshev) norms. Numerical stabilizers use $\varepsilon = 10^{-6}$. Node-deletion bounds are initialized with $\beta_{\min} = 1$ and $\beta_{\max} = \lfloor (HWD)/3 \rfloor$, where $HWD$ is the voxel count of the input volume.

**Reproducibility.** Random seeds are fixed to 42 for Python, NumPy, and PyTorch. CUDA seeds are set via `torch.cuda.manual_seed_all(42)`.

