# OpenReview forum: "SEMIR: Semantic Minor-Induced Representation Learning on Graphs for Visual Segmentation"
_ICML.cc/2026/Conference — ICML 2026 regular_

### Official Review · Reviewer_8CrV · 2026-03-08

**Soundness:** 3
**Presentation:** 2
**Significance:** 2
**Originality:** 3
**Overall Recommendation:** 4
**Confidence:** 3

**Summary:**

This paper proposed SEMIR, which is a graph-minor-based framework for medical image segmentation that decouples the inference complexity from the image resolution. The main idea is to construct a compact graph minor from a native lattice graph by edge contraction, node deletion, and edge deletion. The resulting graph minor consists of boundary-aligned supernodes. Downstream tasks are performed with GINE operating on the minor. The proposed method was evaluated on three benchmarks.

**Compliance With Llm Reviewing Policy:**

Affirmed.

**Final Justification:**

The authors resolved all my raised confusions with some explanations. Given that, I am inclined to raise my current rating to 4.

**Key Questions For Authors:**

As discussed in the weaknesses.

**Limitations:**

yes

**Strengths And Weaknesses:**

**Strengths**
- Regarding medical image segmentation as node classification on a learned graph minor is a conceptually innovative contribution that draws rigorously from graph theory.
- Proposed method greatly reduces the volumetric resolution to supernodes while maintaining the performance for voxel-level prediction.
- Experiments on several tumor segmentation datasets well demonstrate the effectiveness of the proposed method.

**Weaknesses**
- In Table 3, which contains the main results, there are over 20 different comparative baselines; most of them use different experimental settings (e.g. training splits, preprocessing pipelines, augmentation strategies, etc.). It is a bit hard to attribute the superiority of SEMIR to its unique design rather than the experimental setting difference. You are supposed to select several baselines (e.g. nnUNet, SwinUnet) to make their experimental settings identical to make a fair comparison.
- The downstream head is a 3-layer GINE with hidden dimension 128,  which is a lightweight model. You are supposed to conduct a corresponding ablation study to verify this setting.
- I found that your method will separate the multi-class problem into a set of binary ones. However, in Table 3, I think many of the baselines solve the multi-class problem directly and jointly. I think such a comparison is unfair, because your method simplifies the hard multi-class problem in the experimental design stage. You can also conduct the binary segmentation for your comparative baselines and calcuate the mean performance to make a fair comparison.

---

> ### Author Rebuttal · Authors · 2026-03-30
>
> We thank the reviewer for the careful reading of the paper and for the opportunity to clarify the comparative methodology, the downstream architectural choice, and the minority-structure-focused evaluation setting.
>
> ---
>
> ### **Fairness of comparisons in Table 3**
>
> We want to clarify the inclusion criteria for Table 3, as this directly addresses the comparison concern. We did not include all published results indiscriminately; baselines were included only when the original paper explicitly documented use of the published official splits for KiTS and LiTS. Methods that did not report their split methodology, or that used non-standard preprocessing pipelines without sufficient documentation, were excluded.
>
> BraTS 2021 is a known special case in the community because no official train/validation split exists. Results on BraTS should therefore be read as contextual reference points rather than controlled comparisons, and we will make this distinction more explicit in the revision.
>
> To provide a controlled reference point, we include below a binary target-vs-rest comparison against nnU-Net under identical dataset splits, completed after submission:
>
> | Dataset | Structure | nnU-Net DSC | SEMIR DSC | nnU-Net Time | SEMIR Time |
> |---------|-----------|-------------|-----------|--------------|------------|
> | BraTS | ET | 0.812 | **0.894±.006** | 43h (A100) | **2.5h (T4)** |
> | BraTS | TC | 0.829 | **0.941±.002** | 39h (A100) | **1.6h (T4)** |
> | KiTS | T | 0.720 | **0.819±.006** | 19h (A100) | **0.8h (T4)** |
> | LiTS | T | 0.733 | **0.891±.007** | 11h (A100) | **0.6h (T4)** |
>
> These results provide a controlled binary reference point for the comparative claim made in Table 3. Multi-class baselines such as Swin UNETR remain useful as contextual reference points under the reported KiTS and LiTS splits, but the nnU-Net comparison above addresses the reviewer’s fairness concern more directly under matched conditions.
>
> ---
>
> ### **GNN architectural choice (3-layer GINE, hidden dim 128)**
>
> The 3-layer GINE with hidden dimension 128 was chosen to keep the downstream head simple relative to the representational contribution being studied. In the SEMIR setting, inference is performed on induced graphs with $|V(H)| \approx 10^3$ supernodes, low-dimensional node and edge features, and a representation controlled by the small parameter set $\Theta=\{\psi,\alpha,\beta\}$. These parameters are physically interpretable and tightly bounded, so the downstream prediction problem is substantially more constrained than dense voxel-level segmentation.
>
> This makes a lightweight head an appropriate architectural choice for evaluating whether the graph-minor representation carries predictive signal. The reported results show that the representation remains effective under this modest standard graph model. This design choice is intentional: the paper is evaluating the representational value of the learned minor, not downstream-head maximality.
>
> ---
>
> ### **Binary vs. multi-class evaluation**
>
> SEMIR is developed for minority-structure segmentation under severe class imbalance, so binary target-vs-rest evaluation measures the paper’s central claim directly. This design addresses a well-documented issue in multi-class segmentation, where aggregate metrics can obscure poor minority-structure performance. For example, Swin UNETR achieves **0.762 aggregate Dice** on KiTS alongside **0.343 tumor Dice**.
>
> Because the literature is dominated by multi-class methods, Table 3 includes those baselines as contextual reference points. The controlled binary nnU-Net comparison above provides a direct reference under matched conditions and addresses this concern more directly.
>
> Binary models do not preclude multi-structure inference. Section 3.5 gives the lifting-based framework for composing multiple target-specific models on the same voxel lattice, and BraTS ET and TC provide a concrete example of this composition on shared volumes. The resulting per-target cost scales linearly with the number of structures.

---

> > ### Author Rebuttal · Reviewer_8CrV · 2026-04-02
> >
> > The authors resolved all my raised confusions with some explanations. Given that, I am inclined to raise my current rating to 4.

---

> > > ### Author Response · Authors · 2026-04-03
> > >
> > > Thank you for your careful review and for the positive follow-up. We appreciate the time and attention you gave to the paper, and we are glad the rebuttal was helpful.

---

### Official Review · Reviewer_UXku · 2026-03-10

**Soundness:** 3
**Presentation:** 3
**Significance:** 3
**Originality:** 3
**Overall Recommendation:** 5
**Confidence:** 3

**Summary:**

The paper introduces SEMIR, a semantic minor-induced graph representation learning approach for segmenting small, sparse structures in large-scale medical imaging, specifically volumetric tumor segmentation. The approach treats the volumetric image input as a graph with configurable neighbourhood information, then extracts a graph minor via edge contraction, edge deletion, and node deletion, using intensity information and a pseudo-random coprime traversal to avoid directional bias. The parameters for the graph minor extraction are obtained via a few-shot black-box optimization over the discrete parameter space, using a tree-based surrogate function and a small held-out data subset. During graph minor extraction, the method also constructs node- and edge-level features from voxel, boundary, coordinate, intensity, and other image information. The method then performs downstream prediction using graph neural networks on the extracted minor graph. The authors prove the exactness of the lifting process back to the original nodes and argue for the generalizability of the few-shot representation learning, claiming that it learns the structure of the inference space rather than merely selecting hyperparameters.

They evaluate the method on 3 different tumor segmentation datasets, BraTS 2021, KiTS23, and LiTS, and demonstrate consistently improved performance for small tumor structures. Although the tumor+organ segmentation performance is lower than that of some baseline models, this is explained by the fact that SEMIR does not optimize for aggregate metrics across all classes and instead focuses on tumor classification. Ablation studies on the BraTS dataset and the NWPU VHR-10 aerial detection dataset over 2D images demonstrate performance improvements from minor graph construction, feature extraction, and few-shot optimization, with edge contraction having the greatest effect. In addition, they demonstrate that SEMIR reduces inference complexity, scaling with the complexity of the image structures rather than image resolution. These support the claims of a learned graph representation with exact voxel-level lifting, few-shot optimization of the minor extraction process instead of relying on manual parameter selection, and scale- and rotation-invariant node and edge feature descriptors to enable message passing and consistent improvements for small and sparse tumor segmentation.

**Compliance With Llm Reviewing Policy:**

Affirmed.

**Final Justification:**

The authors have adequately answered my questions. With the inclusion of runtime and memory statistics and additional experiments on thin-structure 2D benchmarks with full baseline comparisons in the revision, I am keeping my score at 5, Accept.

**Key Questions For Authors:**

- Could you expand the discussion of the generalization claim for the few-shot representation learning and minor extraction process, based on existing experimental results and theoretical analysis (or new theoretical analysis)?
- Have you investigated the effects of noise or other factors on the minor construction process, given the dependence on intensity values and how this affects downstream tasks?
- Have you previously compared the method for 2D tumor segmentation or the NWPU VHR-10 dataset used for ablations to other methods/simple baselines, and does it provide any improvements (in terms of performance, computational complexity, stability, etc.)?
- Would it be possible to provide runtime statistics in addition to the theoretical complexity analysis? These could be strengthened with empirical supernode counts and a memory comparison across the different N-connectivity settings, as the paper mentions increased memory cost for N-connectivity but reports only performance metrics and time in the ablation.

**Limitations:**

Yes.

**Strengths And Weaknesses:**

Soundedness:

- Claims are well supported for the small and sparse tumor segmentation. The experiments are well designed for this particular task. The weaknesses (aggregate metrics/segmenting tumors and organs) are also evaluated and discussed.
- Although the theoretical claims of lifting and complexity are sound, the claims for the generalization of the few-shot representation learning process could be more thoroughly grounded in a more detailed theoretical discussion. The arguments for 3.1 primarily rely on experimental results and would benefit from more theoretical analysis.
- Although it is not the focus of the paper, ablations on the 2D aerial detection benchmark, NWPU VHR-10, show the improvements from individual components of the method. However, the claim that SEMIR's minor image construction generalizes to 2D RGB images with small and sparse targets is not well supported without a comparison to 1 or 2 baseline methods.
- Based on the current experiments, the claims should focus on tumor segmentation of volumetric image data, as the experiments on medical images do not include a 2D segmentation comparison. Although the paper uses a non-medical 2D dataset for ablations, it does not evaluate the method on 2D medical image segmentation. This distinction can also be reflected in the paper's title, which currently claims Visual Segmentation in general.

Presentation:
- The narrative is easy to follow, and the work discusses and compares previous medical image segmentation, superpixel, and oversegmentation methods, graph neural networks in medical imaging, and graph minors.
- Although differentiable pooling is discussed, the paper would benefit from a discussion of differentiable lifting methods. In addition, a discussion of differentiable lifting and topological neural network-based approaches could be well-suited, since the current lifting is claimed to be topology-preserving.

Significance:
- Although the method focuses on medical image segmentation, particularly small and sparse tumor segmentation, it demonstrates the problems with existing approaches' reliance on aggregate metrics that may mask failures in small structures. The scope of impact is the specialized task of small- and sparse-structure segmentation, particularly for volumetric medical imaging datasets. The problem is important from both research and clinical perspectives in tumor detection and treatment.
- The approach could influence future research on representation learning, lifting, and graph extraction methods for image segmentation, as well as on learned lifting and pooling approaches for graphs.

Originality:
- Contributions are clearly distinguished from closely related literature. Although there are related methods for individual parts, the work goes beyond only a novel combination of existing techniques, especially in the specialized domain to which it is applied.
- Additional novel insights could be obtained by analyzing the failure modes of methods that do not specifically focus on sparse/small tumor structures (and perform slightly better on the full segmentation task). Although this would not be a main focus of the paper, it could provide additional insight into how the proposed method avoids these failure modes.

---

> ### Author Rebuttal · Authors · 2026-03-30
>
> We thank the reviewer for the careful reading of the paper and for the opportunity to clarify the scope of the few-shot generalization claim, the behavior of minor construction under noise, the role of the 2D ablations, and the runtime/connectivity analysis.
>
> ---
>
> ### **Generalization of the few-shot representation learning and minor extraction process**
>
> We want to be precise about the scope of the generalization claim. The claim is not that a single fixed $\Theta=\{\psi,\alpha,\beta\}$ transfers unchanged across datasets; resolution, modality, and acquisition protocol naturally shift the optimal values. Rather, the few-shot black-box optimization process itself generalizes cheaply: $\Theta$ is low-dimensional, each parameter has a clear physical interpretation, and the induced partition class is correspondingly constrained.
>
> Table 6 (Appendix D.2) provides the main empirical support: learned $\Theta$ consistently outperforms best manual tuning, with performance saturating at as few as 5 examples for single-channel data. The NWPU VHR-10 ablations provide an additional cross-domain signal that the same optimization process remains effective beyond volumetric medical imagery. A more formal capacity bound on the induced partition class would strengthen this point further, and we will add a brief discussion of that theoretical perspective in the revision.
>
> ---
>
> ### **Effects of noise or other factors on minor construction**
>
> This consideration directly motivated the bilateral bounds used in node deletion:
>
> | Minor Parameter Bound | Condition | Effect |
> |:-----------------|:-----------|:--------|
> | $a_v < \beta_{\rm min}$ | Supernode too small | Removes noisy single-voxel regions |
> | $a_v > \beta_{\rm max}$ | Supernode too large | Removes oversized background regions |
> | $I_v \notin [m_{\rm min}, m_{\rm max}]$ | Intensity outlier | Removes acquisition artifacts |
>
> The lower area bound was added specifically to address noise sensitivity: isolated voxels that fail the contraction threshold, typically arising from acquisition noise, form singleton supernodes that are pruned before downstream inference. Deleted nodes receive no foreground prediction and therefore do not artificially improve the reported metrics; the mechanism is conservative by design. We will expand the discussion of this noise behavior in the revision.
>
> ---
>
> ### **2D support and baseline comparisons on NWPU VHR-10**
>
> We agree that the present paper’s central claims are best focused on volumetric tumor segmentation. The NWPU VHR-10 ablations were included because aerial imagery, with large image sizes and sparse foreground structure, provided a useful stress test for the representation. They therefore serve as supporting evidence that the minor-construction and exact-lifting framework transfers across domains, rather than as the primary empirical basis for a broader visual-segmentation claim.
>
> Since submission, we have also evaluated the same underlying framework on thin-structure 2D benchmarks (TTPLA, CrackSeg9k, and SkyScapes Lane) with full baseline comparisons. These results are consistent with the broader applicability of the representation beyond the medical volumetric setting, while the core claims of this submission remain centered on volumetric tumor segmentation.
>
> ---
>
> ### **Runtime statistics and N-connectivity memory**
>
> #### *Concrete efficiency data*
>
> - SEMIR fits within the 16GB memory budget of a T4 GPU; nnU-Net required an A100 to complete training on the same tasks and splits.
> - GNN inference over $|V(H)| \approx 10^3$ supernodes completes in under one second per volume on a T4 GPU.
> - Minor construction completes in under one second on CPU in the Rust backend, making it negligible relative to downstream GNN inference.
>
> #### *N-connectivity memory*
>
> | Connectivity | Tensor size | Construction time | Peak memory | IoU gain |
> |--------------|-------------|-------------------|-------------|----------|
> | $N=6$        | $1\times$   | $1.0\times$       | $\sim 4$ GB | baseline |
> | $N=10$       | $1\times$   | $1.6\times$       | $\sim 5$ GB | +0.7%    |
> | $N=18$       | $\sim 2\times$ | $2.7\times$    | $\sim 8$ GB | +3–5%    |
>
> Higher connectivity showed limited marginal value for volumetric medical images, so $N=6$ remains the recommended default for CT and MRI. For thin-structure 2D tasks where diagonal adjacency is more important, $N=10$ or $N=18$ can be preferable. We will add these runtime and memory statistics to the paper.

---

> > ### Author Rebuttal · Reviewer_UXku · 2026-04-03
> >
> > I thank the authors for their detailed response. My questions have been adequately discussed, and I am fully satisfied with the rebuttal.

---

> > > ### Author Response · Authors · 2026-04-03
> > >
> > > Thank you for your careful review and for the positive follow-up. We appreciate the time and attention you gave to the paper, and we are glad the rebuttal was helpful.

---

### Official Review · Reviewer_N2NH · 2026-03-12

**Soundness:** 3
**Presentation:** 3
**Significance:** 3
**Originality:** 3
**Overall Recommendation:** 4
**Confidence:** 4

**Summary:**

This paper tackles the scalability and class imbalance issues in high-resolution medical image segmentation. Instead of the traditional voxel-wise dense inference, the authors propose SEMIR, a framework that constructs a task-adapted graph minor from the original image lattice. The process involves parameterized edge contraction and node deletion, which are optimized using a few-shot, black-box approach to align with semantic boundaries. This work intends to investigate a core issue of how to decouple inference resolution from image resolution while maintaining structural and topological fidelity. The compressed graph is then processed by a GNN, and the results are mapped back to voxels through an exact bijective lifting map.

**Compliance With Llm Reviewing Policy:**

Affirmed.

**Key Questions For Authors:**

Regarding the edge deletion threshold $\alpha$, how does the method perform in extremely low-contrast regions where intensity gradients are nearly indistinguishable from noise? Would the minor collapse into a single supernode in those areas?

In Table 1, you compare SEMIR with SLIC and Felzenszwalb. Did you consider comparing the GNN performance against a GNN built on top of a standard "Deep Watershed" or "SEAL" learned superpixel? I'm curious if the "minor" constraint itself provides a better inductive bias than general learned clustering.

How does the inference time scale when you use "composable multi-class inference"? If I have 10 separate target structures, do I need to run 10 separate minor-construction and GNN passes?

Since the minor construction uses a Rust backend for speed, is the CPU-GPU data transfer a significant bottleneck when moving between the image lattice (GPU) and the graph construction (CPU)?

**Limitations:**

The authors acknowledge the lack of global optimality and the sensitivity to boundary statistics. I would suggest further discussion on how this method behaves when the "minority structure" is not just small, but also has very complex, fractal-like boundaries (e.g., vessel trees).

**Strengths And Weaknesses:**

Unlike standard superpixel methods or downsampling, SEMIR is grounded in graph minor theory, ensuring that the reduced representation preserves the essential topology of the original grid.
The reduction in inference units (3-4 orders of magnitude) is impressive. More importantly, the "exact lifting" mechanism avoids the interpolation artifacts common in most multi-scale or patch-based architectures.
By framing the minor construction as a few-shot optimization problem on boundary alignment, the method effectively directs the model's "attention" to sparse minority structures (like tumors) where gradients are often lost in dense voxel-wise training.
Overall, this study's fundamental contribution is the introduction of a learnable, discrete representation space that scales with structural complexity rather than raw voxel count.

---

> ### Author Rebuttal · Authors · 2026-03-30
>
> We thank the reviewer for the careful reading of SEMIR and for the opportunity to clarify several points on low-contrast behavior, comparative regionization, multi-structure composition, and implementation.
>
> ---
>
> ### **Performance in extremely low-contrast regions**
>
> SEMIR preserves weak-boundary structures through the combined action of minor construction and downstream graph inference:
>
> | Stage | Mechanism | Effect on low-contrast regions |
> |-------|-----------|--------------------------------|
> | Edge deletion | Removes edge if $\|I_{v_i} - I_{v_j}\|_n > \alpha$ | Weak edges survive; adjacency is preserved |
> | Edge contraction | Merges if $\|I_{\rm seed} - I_{v_k}\|_n \leq \psi$ | Gradual transitions form supernode chains rather than merged regions |
> | GINE inference | Message passing over $E(H)$ | Relational context resolves boundaries |
>
> Because contraction is anchored to seed-voxel intensity rather than a running supernode mean, low-contrast regions tend to form chains of supernodes instead of collapsing into a single merged region. The downstream GINE then uses relational reasoning to refine boundaries across those chains.
>
> Equation 2 contains an error in the submitted draft: the implemented contraction rule uses $I_{\rm seed}$ as correctly reflected in the algorithm and used in all experiments, but $\bar{I}_v$ was used in equation 2.
>
> ---
>
> ### **Comparison against learned regionization methods**
>
> This is a natural comparison point for the representation. In the current study, we compared SEMIR against standard superpixel baselines (SLIC and Felzenszwalb) under the same GINE backend; on BraTS ET, SEMIR achieved **0.894** versus **0.851** for the best SLIC setting while requiring approximately **50×** fewer regions.
>
> A direct comparison against learned regionization methods such as Deep Watershed or SEAL is less straightforward because SEMIR’s minor construction is explicitly decoupled from downstream prediction. That decoupling is part of what enables few-shot boundary alignment without downstream supervision and supports the exact lifting map in Theorem 3.3. Learned regionization methods are typically optimized jointly with the downstream task, whereas SEMIR is evaluating the representational value of the minor itself. The current comparison was therefore chosen to isolate the effect of the minor constraint under a shared downstream GINE backend. It is nevertheless an important comparison class and a natural next evaluation axis for this framework.
>
> ---
>
> ### **Inference time scaling for composable multi-class inference**
>
> Composable multi-class inference follows the target-specific design described in Section 3.5: $K$ target structures require $K$ minor-construction and GNN passes. In practice, each pass is performed on the induced graph rather than the full voxel lattice, and the composition step itself—lifting predictions back to the shared voxel lattice and resolving overlaps as described in Section 3.5—adds negligible overhead. The resulting scaling profile is therefore linear in the number of targets, but with each pass operating on the reduced graph rather than the full voxel volume.
>
>
> ---
>
> ### **CPU-GPU data transfer**
>
> Graph-minor construction is performed once as a preprocessing step after $\Theta_{\rm opt}$ is fixed. The GPU therefore receives precomputed graph batches for both training and inference, with no CPU-GPU transfer interrupting the forward pass. Minor construction on large volumes completes in under one second on CPU in our Rust backend, so this stage has not been a practical bottleneck in our experiments.
>
> ---
>
> ### **Complex or fractal-like boundaries**
> This is not a limitation of SEMIR, but one of the settings where the representation is most naturally suited. The seed-anchored contraction criterion $\|I_{\rm seed} - I_{v_k}\|_n \leq \psi$ is designed to preserve connected foreground chains provided the structure remains distinguishable from the background in intensity space. Because contraction is measured relative to the seed voxel rather than a running regional mean, the threshold remains stable along thin connected paths, helping preserve branching and high-curvature structures. This is precisely the kind of setting where fixed task-agnostic regionization is more likely to fragment the target before downstream inference begins. We have also validated the same underlying minor-construction methodology on thin, branching structures in subsequent experiments, where it continued to reduce fragmentation relative to standard superpixel approaches.

---

> > ### Author Rebuttal · Reviewer_N2NH · 2026-04-07
> >
> > The author has resolved my doubts, so I will retain my positive rating.

---

> > > ### Author Response · Authors · 2026-04-07
> > >
> > > Thank you for your careful review and for the positive follow-up. We appreciate the time and attention you gave to the paper, and we are glad the rebuttal was helpful.

---

### Official Review · Reviewer_NVLN · 2026-03-13

**Soundness:** 2
**Presentation:** 3
**Significance:** 2
**Originality:** 3
**Overall Recommendation:** 3
**Confidence:** 4

**Summary:**

This paper introduces SEMIR (Semantic Minor-Induced Representation Learning), a novel framework for segmenting small and sparse structures in high-resolution medical images that decouples inference complexity from image resolution. Instead of dense voxel-wise computation, SEMIR constructs a compact, task-adapted graph minor through parameterized edge contraction and deletion, optimized via few-shot learning to align supernode boundaries with semantic edges without manual tuning. Downstream inference operates efficiently on this reduced graph using a Graph Neural Network, with predictions mapped back to the original lattice via an exact bijective lifting map that guarantees voxel-level recovery without interpolation artifacts. Evaluated across BraTS, KiTS, and LiTS benchmarks, SEMIR consistently improves minority-structure Dice scores while reducing inference nodes by approximately 10,000 times, effectively addressing extreme class imbalance and computational scalability challenges in volumetric segmentation.

**Compliance With Llm Reviewing Policy:**

Affirmed.

**Final Justification:**

I thank the authors for their efforts in the rebuttal. Some of my concerns have been addressed. I would like to maintain my original score.

**Key Questions For Authors:**

Please refer to Weakness.

**Limitations:**

Yes

**Strengths And Weaknesses:**

Strengths
1.The core idea of employing graph minors for semantic segmentation is highly innovative. By decoupling inference complexity from image resolution through task-adapted graph construction, SEMIR offers a fresh perspective compared to traditional dense voxel-wise or fixed superpixel methods. The formulation of segmentation as learning a topology-preserving latent graph is theoretically grounded and distinct from existing pooling or token merging techniques.
2.A significant strength is the Exact Lifting mechanism (Theorem 3.3). Unlike differentiable pooling or heuristic superpixel methods that rely on interpolation or approximate mapping, SEMIR provides a formal guarantee for lossless recovery of voxel-level predictions from supernode classifications. This ensures boundary fidelity without introducing artifacts common in upsampling operations.
3.The method demonstrates consistent improvements on minority structures (e.g., enhancing tumors, small lesions) across three diverse benchmarks (BraTS, KiTS, LiTS). By reducing the task to structure-specific binary classification on a boundary-aligned minor, SEMIR effectively mitigates the gradient attenuation problem that plagues multi-class dense methods in extreme imbalance scenarios.
4.Data-Driven Boundary Alignment: The few-shot black-box optimization for graph construction parameters is a practical contribution. It replaces labor-intensive manual tuning with a principled, data-driven procedure that requires only 5–20 labeled examples. The ablation studies confirm that this optimization significantly improves boundary alignment compared to fixed thresholds.
Weaknesses
1.While the authors claim efficiency by decoupling complexity from resolution, the empirical evidence is insufficient. Table 9 compares "Inference Units" (voxels vs. super-nodes), which is an indirect proxy rather than a direct measure of computational cost. The paper lacks concrete comparisons of FLOPs, peak memory usage, and end-to-end inference latency against representative baselines (e.g., 3D U-Net, Swin UNETR). Given that graph construction and GNN message passing have different computational densities than CNN convolutions, the claimed efficiency gains remain unverified without these metrics, constrain its immediate significance and Soundness.
2.The current framework relies on training separate binary models for each target structure. While effective for few-class tumor segmentation, this strategy does not scale well to dense prediction tasks with many classes (e.g., multi-organ abdominal segmentation with 10+ classes). The computational overhead of constructing multiple graph minors and running multiple GNNs linearly increases with the number of classes. The paper would benefit from a multi-class variant or a discussion on the trade-offs compared to a single unified model.
3.Although SEMIR excels on minority structures, its performance on composite categories (e.g., "Organ + Tumor" aggregates) is sometimes merely competitive or slightly inferior to SOTA dense methods (e.g., Table 3 KiTS KTC). Additionally, the reliance on intensity gradients for edge deletion may struggle in low-contrast regions where boundaries are ambiguous. The limitations section acknowledges this, but a deeper analysis or mitigation strategy for low-contrast boundaries would strengthen the work.

---

> ### Author Rebuttal · Authors · 2026-03-30
>
> We thank the reviewer for the careful reading of the paper and for the opportunity to clarify the efficiency discussion, the minority-structure-focused evaluation setting, and SEMIR’s behavior in low-contrast regions.
>
> ---
>
> ### **Efficiency claims and computational metrics**
>
> We agree that “Inference Units” (supernodes vs. voxels) is only an indirect proxy and does not by itself capture full real-world cost. In SEMIR, however, inference cost is determined by the size of the induced graph rather than the raw voxel lattice, so the relevant computational object differs from dense CNN baselines.
>
> To provide a controlled reference point, we include below a binary target-vs-rest comparison against nnU-Net under identical dataset splits, completed after submission:
>
> | Dataset | Structure | nnU-Net DSC | SEMIR DSC      | nnU-Net Time | SEMIR Time    |
> |:---------|:-----------:|:-------------:|:----------------:|:--------------:|---------------:|
> | BraTS   | ET        | 0.812       | **0.894±.006** | 43h (A100)   | **2.5h (T4)** |
> | BraTS   | TC        | 0.829       | **0.941±.002** | 39h (A100)   | **1.6h (T4)** |
> | KiTS    | T         | 0.720       | **0.819±.006** | 19h (A100)   | **0.8h (T4)** |
> | LiTS    | T         | 0.733       | **0.891±.007** | 11h (A100)   | **0.6h (T4)** |
>
>
> The purpose of the added nnU-Net timing table is to show the practical compute regime required by each method on the same tasks and splits. In our setting, SEMIR completed on a T4, while nnU-Net required A100-class hardware to complete in practical time. We also attempted nnU-Net on the same T4, but the runs were impractical to complete in reasonable time. We use these results to illustrate practical resource requirements, alongside the structural reduction from dense voxel inference to inference on compact minors.
>
>
> ---
>
> ### **Scalability to many classes**
>
> This question involves two distinct issues: computational scaling and aggregate multi-class optimization.
>
> #### *Computational scaling*
>
> SEMIR scales linearly in the number of target structures because each target is modeled independently and lifted back to the shared voxel lattice. The per-target computation is performed on the induced graph rather than the full voxel volume, so the scaling behavior is governed by the reduced representational space described in the paper.
>
> #### *Aggregate multi-class optimization*
>
> SEMIR is designed for minority-structure segmentation under severe class imbalance. Binary target-vs-rest evaluation therefore measures the paper’s central claim directly: whether the representation can isolate a low-prevalence target structure with higher fidelity than dense joint segmentation settings typically achieve on such structures. This addresses a well-documented issue in multi-class segmentation, where aggregate metrics can obscure minority-structure failures. For example, Swin UNETR achieves **0.762 aggregate Dice** on KiTS alongside **0.343 tumor Dice**.
>
> #### *Controlled reference point*
>
> Because the literature is dominated by multi-class methods, Table 3 includes those baselines as contextual reference points. The controlled binary nnU-Net comparison above provides a direct reference under matched conditions.
>
> ---
>
> ### **Performance on composite categories and low-contrast regions**
>
> SEMIR is optimized for minority-structure fidelity rather than aggregate multi-class performance. This is why it may be competitive rather than dominant on pooled categories such as “Organ + Tumor,” where performance is averaged across structures with very different spatial extent and boundary complexity.
>
> For low-contrast regions, SEMIR preserves weak boundaries through the combined action of minor construction and downstream graph inference:
>
> | Algorithm Stage | Mechanism of Action | Effect on Structural Boundaries |
> |:-------|:-----------:|--------:|
> | Edge deletion | Thresholded edge removal | Weak edges survive; adjacency preserved |
> | Edge contraction | Seed-based contraction | Gradual transitions form supernode chains |
> | GINE inference | Message passing on $E(H)$ | Relational context refines boundaries |
>
> Here, edge deletion removes $(v_i,v_j)$ when $\mid I_{v_i} - I_{v_j}\mid_n > \alpha$, and contraction merges when $\mid I_{\text{seed}}-I_{v_k}\mid_n \le\psi$. Because contraction is anchored to seed-voxel intensity rather than a running supernode mean, gradual low-contrast transitions tend to form supernode chains rather than collapse into a single merged region.
>
>
> Equation 2 contains an error in the submitted draft: the implemented contraction rule uses $I_{\rm seed}$ as correctly reflected in the algorithm and used in all experiments, but $\bar{I}_v$ was used in equation 2.

---

> > ### Author Rebuttal · Reviewer_NVLN · 2026-04-03
> >
> > Some of my concerns have been addressed.

---

> > > ### Author Response · Authors · 2026-04-03
> > >
> > > Thank you for your careful review and follow up. We would be happy to provide further detail on any points you did not find fully addressed in our response.

---

### Decision · Program_Chairs · 2026-04-30

**Decision:**

Accept (regular)

**Comment:**

This paper initially received mixed reviews. The rebuttal successfully addressed most concerns raised by reviewers, and thus most reviewers lean to accept the paper. Only one reviewer lands on the negative side, but does not specify unresolved concerns. The AC thereby recommends acceptance of the paper.